# Agent Simulation Model of COVID-19 Epidemic Agent-Based on GIS: A Case Study of Huangpu District, Shanghai

**DOI:** 10.3390/ijerph191610242

**Published:** 2022-08-18

**Authors:** Tao Dong, Wen Dong, Quanli Xu

**Affiliations:** 1School of Information Science and Technology, Yunnan Normal University, Kunming 650500, China; 2Faculty of Geography, Yunnan Normal University, Kunming 650500, China; 3GIS Technology Engineering Research Centre for West-China Resources and Environment of Educational Ministry, Yunnan Normal University, Kunming 650500, China

**Keywords:** COVID-19, agent-based simulation model, population mobility, lockdown intervention

## Abstract

Since the COVID-19 outbreak was detected and reported at the end of 2019, the pandemic continues worldwide, with public health authorities and the general public in each country struggling to balance safety and normal travel activities. However, the complex public health environment and the complexity of human behaviors, as well as the constant mutation of the COVID-19 virus, requires the development of theoretical and simulation tools to accurately model all segments of society. In this paper, an agent-based model is proposed, the model constructs the real geographical environment of Shanghai Huangpu District based on the building statistics data of Shanghai Huangpu District, and the real population data of Shanghai Huangpu District based on the data of China’s seventh Population census in 2020. After incorporating the detailed elements of COVID-19 transmission and the real data of WHO, the model forms various impact parameters. Finally, the model was validated according to the COVID-19 data reported by the official, and the model is applied to a hypothetical scenario. Shanghai is one of the places hardest hit by the current outbreak, Huangpu District is the “heart, window and name card” of Shanghai, and its importance to Shanghai is self-evident. so we used one-to-one population modeling to simulate the spread of COVID-19 in Huangpu District of Shanghai, In addition to the conventional functions of crowd movement, detection and treatment, the model also takes into account the burden of nucleic acid detection on the model caused by diseases similar to COVID-19, such as seasonal cold. The model validation results show that we have constructed a COVID-19 epidemic agent risk assessment system suitable for the individual epidemiological characteristics of COVID-19 in China, which can adjust and reflect on the existing COVID-19 epidemic intervention strategies and individual health behaviors. To provide scientific theoretical basis and information decision-making tools for effective prevention and control of COVID-19 and public health intervention in China.

## 1. Introduction

Shortly after the first COVID-19 case was detected and reported in Wuhan, Hubei Province, China, in late December 2019, the COVID-19 virus quickly spread to many countries around the world [1]. As a result, the World Health Organization (WHO) declared the situation a Public Health Emergency of International concern on 30 January 2020 and subsequently designated it a pandemic on 11 March 2020 [2]. As of 13 June 2022, WHO has reported 532,887,351 confirmed cases of COVID-19, while 6,307,021 deaths from COVID-19 have been reported [3]. As of 13 June 2022, a total of 3,496,075 confirmed COVID-19 cases, including 18,896 deaths, had been reported on the official website of The National Health Commission of China. COVID-19 is still sporadic in China, with cases increasing rapidly in some regions, and the prevention and control situation remains complex. Although China has entered the stage of normalizing the prevention and control of COVID-19, risk factors and intervention management are still hot spots. China is still seeking a delicate balance between the safety of people and normal travel activities, aiming to enable people to safely resume normal travel and work. Effective testing and timely contact tracing are of great significance to the effective prevention and control of COVID-19 and the rational development of public health projections in China, as well as to reduce the spread of COVID-19 [4,5,6,7]. While important issues regarding testing are identifying infected people and their contacts, addressing these issues requires a better understanding of community structures, COVID-19 outbreak locations and individual lifestyles [8,9]. However, the scale of the COVID-19 outbreak and its typical clinical symptoms (fever, dry cough, myalgia and pneumonia) are strikingly similar to those of influenza [10].This will place a great burden on the medical resources for testing, and the additional burden on traditional testing sites (hospitals, emergency rooms, etc.) poses a great challenge to the safety of these sites [11,12,13]. The development and analysis of mathematical models play an important role in the control and prevention of disease transmission. The Compartment Model is the foundation for understanding the complex dynamics of epidemics and a powerful mathematical framework [14]. At the same time, the Compartment Model is an effective tool for understanding epidemics and evaluating potential countermeasures [15,16,17,18,19]. The agent-based model is a class of Compartment Model that provides a highly accurate representation of time and space at the personal level [5,20,21,22,23,24]. The agent model considers multiple types of physical locations, such as businesses, homes, schools, hospitals and nursing homes, and analyzes local human behavior trends and population movement patterns from the unique characteristics of the community.

Once validated, the agent model can be used to test competing “what-if” scenarios that would otherwise be impractical and unwarranted experiments [25]. For example, Ferguson et al. [26] used census data to generate population age and family distribution in the agent model, and used commuting distance survey data to simulate individual workplaces in the model to predict the number of deaths in the UK under different epidemic prevention strategies. Its strategies include home isolation for the infected, home isolation for family members, social distancing for people over 70, social distancing for all individuals, and closing schools and universities. The other part of the research combines the real trajectory data of the individual in the agent model to increase the reality of the spatio-temporal behavior of the individual in the model. Aleta et al. [5] used POI(Point of Interest) access data of anonymous mobile phone users to construct a weighted contact network between individuals in the agent model to simulate the impact of different contact tracing, home quarantine and detection ratios on the second wave of COVID-19 in Boston. Koo et al. [27] constructed an agent model based on Singapore population census data and individual public transport card swipe data, and estimated the transmission of SARS-CoV-2 under three different R0 values. The cumulative number of infected persons under four different intervention schemes (isolation of patients and family members, isolation of patients and family members and closure of school for 2 weeks, isolation of patients and family members and encouragement to work from home, isolation of patients and family members and closure of school and encouragement to work from home) were simulated. Müller et al. [28] simulated the activity intensity of individuals in the agent model over time based on the location data of mobile phones, revealing the impact of reducing contact between individuals, closing schools, and shifting leisure activities from outdoor to indoor on the development of the epidemic in Berlin. Gressman and Peck et al. [29] studied an agent model based on university campus to study the strategy of university resumption under the epidemic situation. Hinch et al. [30] developed an agent-based open source modeling framework to support non-pharmaceutical interventions and analysis of population contact tracing protocols. Yin et al. [31] combination of mobile location, census, the census and building the trip survey such as multi-source space-time data to construct must level of the individual agent model, simulate COVID-19 in Shenzhen propagation process, systematic evaluation of the family, work, public contacts isolation, wore effect on inhibition of epidemic situation, timely detection. A combination of non-drug interventions was recommended for the normalized epidemic management in cities with different epidemic prevention needs. Therefore, agent-based infectious disease models, especially spatial explicit models that can clearly express the location/place of individual movement and activity, can effectively support interventions targeted at different groups, places and geographical locations. Truszkowska A. et al. [25] took New Rochelle, USA, as an example to build an agent model and study the effects of different vaccination strategies on epidemic prevention and control. Hanisha Tatapudi et al. [32] studied the impact of vaccine priority strategies on mitigation of COVID-19 by constructing an agent model based on demographic and social data of an urban community with 2.8 million inhabitants in the United States. The advantages of the agent model have been proved by a large number of experimental studies, confirming its realizability in technical implementation and its extensibility in cross-scenario research [33,34,35,36,37]. Existing agent models focus on either a very small microenvironment (such as a campus) or the environment of the entire country for agent simulation, and deliberately coarsening the population for numerical simulation. Medium-scale agent models constitute an important but unconsidered modeling scale for the COVID-19 epidemic. The highly accurate agent model will closely capture real-world community and interaction patterns on this medium-scale model, so that the model can reflect the specific lifestyle of China in detail. At the same time, it can avoid coarsening the population to simulate large-scale activities. Therefore, in order to build a high-precision agent model, we selected Huangpu District of Shanghai, which is relatively serious in the current epidemic, to build an agent model based on Huangpu District of Shanghai. Shuli Zhou et al. [38] based on the existing mobile network operator data and gravity model, presented the transmission process of COVID-19 well and identified four transmission modes, which are highly dependent on the urban spatial structure and location. Therefore, our agent model conducts one-to-one simulation of the geographical location and population data of Huangpu District in Shanghai, which fully reflects the geographical topography of Huangpu District and the data structure of the national census. The one-to-one generation of virtual population and the one-to-one generation of the residential and public buildings of the population opens up new possibilities for the analysis of epidemiology. This is of great significance to the effective prevention and control of COVID-19 in China and the development of targeted public health dry prediction. This model is based on the early agent model used by Ferguson et al. [21,23,24] to study pandemic influenza, and at the same time, the function of the model is improved by referring to the agent-based simulation study of American urban areas created by Hanisha Tatapudi et al. [32]. Our model includes the isolation of different areas in Huangpu District and the isolation of the whole Huangpu District, as well as the nucleic acid testing practice (hospital testing, home testing), treatment methods (such as general hospitalization, ICU hospitalization, and home isolation after treatment). At the same time, because the symptoms of diseases such as seasonal influenza or the common cold are similar to those of COVID-19, these patients will put an additional burden on testing resources and play an important role during the COVID-19 epidemic. Our model will target these patients to make the model more realistic. The study of Junfeng Jiang et al. [39] showed that population mobility was the driving factor for the rapid spread of COVID-19, so we constructed a population mobility model to simulate the spread of COVID-19 in Huangpu District of Shanghai. At the same time, due to the particularity of schools, hospitals and nursing homes, our model carries out separate modeling for employees in these three places, so as to specially consider the particularity of these professions. Finally, our model can selectively study the government’s macro-intervention measures, such as the partial closure and isolation of Huangpu District and the whole closure and isolation of Huangpu District, as well as the reopening of these places.

Our model is guided by the epidemiological characteristics of COVID-19 and the agent modeling method, and based on the interaction mechanism between the risk of COVID-19 outbreak, individual epidemiological influencing factors and macro-intervention behavior, and the optimization of the algorithm in the agent modeling. Taking the risk probability of COVID-19 outbreak in China as the analysis and evaluation index, a COVID-19 epidemic agent disease risk prediction model was constructed to simulate and predict the dynamic evolution of COVID-19 outbreak risk from micro to macro, combining individual epidemiological influencing factors and macro intervention behaviors. Finally, the COVID-19 epidemic agent risk assessment system suitable for the individual epidemiological characteristics of COVID-19 in China will be constructed, and universal COVID-19 intervention and regulation means and indicators will be explored, in order to adjust and reflect on existing COVID-19 intervention strategies and individual health behaviors. To provide scientific theoretical basis and information-based decision-making tools for effective COVID-19 prevention and control and public health intervention in China.

## 2. Materials and Methods

### 2.1. Data Sources

We collected and sorted out the geographical coordinates, building types and capacity of all kinds of buildings in Huangpu District of Shanghai, including residential buildings, hospitals, schools, nursing homes, public places and leisure places. These data were obtained through the API interface of Amap [40] and supplemented by comparison with manual collection of Google Maps [41]. The population data was from the seventh population census of China in 2020. The population model is created by using the data of the seventh population census of Huangpu District, Shanghai. The number of students and staff in schools, the number of staff in hospitals and the number of patients in hospitals are all from the statistics of Education Bureau and Health Bureau of Huangpu District. Figure 1 shows where the model is created.

Based on the statistics of the seventh Population census of China, the number of existing families, the average population of a family and the housing vacancy rate in Huangpu District, we allocated houses to agents, creating 265,425 families and 662,030 agents in total. Figure 2a is the population distribution of Huangpu District at the beginning of the model, and Figure 2b is the distribution of each age group accurately generated by the model according to the census data. Based on census data, we divided people over the age of 60 into family and nursing home residents. This is particularly important for realistic predictions of COVID-19 virus disease, which is more severe and deadly in older people. All intelligents aged 5–17 are assigned to all kindergartens, middle schools and high schools, and the number of students is allocated according to the size of the school. Similarly, both universities and vocational colleges assign agent aged 18–24 according to the size of the school. Workplace distribution is generated from occupational statistics, along with the number of workers in hospitals, schools and nursing homes based on their size.

The National Health Commission of the People’s Republic of China (NHC) is the most authoritative health supervision and management body in China. After the outbreak of COVID-19, national and provincial health commissions issued daily notifications on COVID-19 in the form of articles. This includes data on new and cumulative COVID-19 confirmed cases and deaths. The epidemic data used in this study were from the National Health Commission of the People’s Republic of China [42] and Shanghai Municipal Health Commission [43]. Data is obtained and analyzed by Python web crawler, and missing data is supplemented by network platform. Death data, we collected the death rate and test data of the whole Shanghai city, and then calculated the death data of Huangpu District according to the proportion of population.

### 2.2. Model Overview

The overall structure of the model is shown in Figure 3: We will build the agent model from the three aspects in the figure, using building statistics to create a map of Huangpu District in Shanghai, using census data to create individuals, and finally combining relevant parameters and crowd movement model to form the agent model.

Our model according to the statistics of Huangpu District building, obtain the building type and name, and then according to the structure type and name from Scott maps API interface to get the latitude and longitude of the buildings according to the longitude and latitude to generate Huangpu District building, resulting in Huangpu District maps, as far as possible, the reduction of the region’s geographical environment. According to the census data of Huangpu District and family size and family structure data, access to research areas of resident population, age accounted for the total number of registered permanent residence, head of the household, age distribution, the average family size or childless families, single-parent families and families with elderly proportion has an obvious local community structure of the data. Firstly, the families in Huangpu District were created according to these data, and then family members were created based on the families. One-to-one numerical modeling was conducted for local residents as much as possible, and local community characteristics were restored as much as possible. Starting from the unique characteristics of the community, the local human behavior trend and local population movement pattern are analyzed in combination with the total number of people working and the proportion of the means of transportation they take to work. When residents live at home, they are combined with the transmission rate of THE COVID-19 virus at home, the transmission rate of the COVID-19 virus in the environment during the movement of people, and the transmission rate of the COVID-19 virus in workplaces, schools and hospitals. Diagnose patients can according to the requirements of the parameters such as the age is divided into general hospitalized patients and ICU patients, diagnosed patients treated in hospital, combination of COVID-19 parameters such as recovery time for recovery in patients with confirmed, need ICU patients, combination of treatment in the ICU mortality and age related parameters to predict the number of deaths in patients with confirmed. Confirmed patients who recover or die have no further impact on the spread of COVID-19, thus forming a complete agent model. Our model is designed to be based on a simulated spatial environment highly associated with the real geographical environment, which is a seamless integration of various agent objects and geographical spatial environment.

Model SEIR is a Compartment Model. Since the SEIR epidemic model proposed and studied by Cooke and Driessche [44], it has become a classical model for the study of epidemics and the most important model for disease control. Ferguson et al. [23,24,26] investigated the impact of non-pharmaceutical interventions on the spread of COVID-19 based on the agent model developed by SEIR model. Our model on the basis of the extension, is composed of five kinds of state: Susceptible (S) is not sick, but the lack of immune ability, after contact with an infected person vulnerable to infection; Exposed (E) refers to come into contact with infected people, but no ability to infect others, the long incubation period for infectious diseases, however, due to the national health committee of the People’s Republic of China announced that 2019 coronary virus disease of the incubation period is about 10 days, the incubation period is contagious [42],so our model in the incubation period of infectious; Infected (I) refers to a person infected with an infectious disease, which may be transmitted to a category S member to make him or her a Category E or I member; Removed (R) or Dead (D) refers to a person isolated or immune from illness or who has died after treatment. Removed or Dead an infected person is no longer a factor in the spread of COVID-19. As showed in Figure 4, the agent’s state transitions and some extensions to the SEIR model are described in detail.

At each simulation step (Δ*t*), each agent switches between the locations where the model is generated (home, workplace, leisure, school, hospital, and nursing home). In these sites, agents interact with transmission parameters of the COVID-19 virus in these sites, thereby allowing the COVID-19 virus to spread in these sites. Once COVID-19 breaks out, this area will be sealed off and isolated. Therefore, in our model, we assume Huangpu District is isolated. Therefore, these agents cannot leave the simulated area during this period of time, and new agents cannot enter the area during the simulation.

According to Ferguson et al. [23,24], there is no difference between day and night in a day. In the model, the COVID-19 virus interacts with the environmental transmission rate through the location of the agent to constitute the transmission of COVID-19 in the model. At the same time, due to the particularity of schools, hospitals and nursing homes, we clearly modeled these key roles for these high-risk places. Agents can become infected with COVID-19 disease and cause family members, hospital employees, or other agents in the workplace to become infected with COVID-19 virus. For example, the probability that a worker who works in a public place will contract COVID-19 disease is calculated based on their exposure in the workplace and home.

Our model generates a series of locations (family homes, schools, hospitals, public places, nursing homes, etc.) in Huangpu District of Shanghai *L* = {1,2,3,…,*l*}, and generate the permanent residents of Huangpu District *N* = {1,2,3,…,n}. We defined a set of functions based on the Haversine formula to send the generated residents to the various generated locations: fq:N→L, including *q* ∈ {H,W,S,Rh,Hsp}, function fq will each related agent *I*
∈ *V*
∈
*N* sent to the location of the associated *q*. Types of *q* include home (H), public place (W), school (S), nursing home (Rh), and hospital (Hsp).Since every agent cannot be associated with all types of positions, we use fq(i)=Ø to indicate that agent *i* is independent of type *q*, and $n_{l}$ to indicate the number of all agents associated with position *l*.

The Haversine equation is a method of calculating the distance between two points based on latitude and longitude. We consider various modes of travel (bus, subway, private car, etc.), and the time for an agent to arrive at its destination varies depending on the mode of travel. The Haversine formula is:(1)$$d=2Rarcsin\left(\sqrt{\sin^{2}\left(\frac{lat2-lat1}{2}\right)+\cos\left(lat2\right)\cos\left(lat1\right)\sin^{2}\left(\frac{lon2-lon1}{2}\right)}\right)$$

In each simulation step (Δ*t*), the probability of susceptible (S) agent *i* contracting the COVID-19 virus is related to its location. Specifically, in a simulation step, the probability of agent *i* contracting COVID-19 is:(2)$$P_{i}\left(t\right)=1-e^{\Delta t\tau_{i}\left(t\right)}$$

$\tau_{i}\left(t\right)$ is a nonnegative time-varying parameter that quantifies the infectious risk at all locations associated with agent *i,* i.e., the infectious risk at home, workplace, hospital, etc., and is equal to:(3)$$\tau_{i}\left(t\right)=\lambda_{Hf_{H}\left(i\right)}\left(t\right)+\lambda_{Wf_{W}\left(i\right)}\left(t\right)+\lambda_{Sf_{S}\left(i\right)}\left(t\right)+\lambda_{Rhf_{Rh}\left(i\right)}\left(t\right)+\lambda_{Hspf_{Hsp}\left(i\right)}\left(t\right)$$
where λ represents the infectivity function of each location associated with agent *i*. At the same time, we use $\lambda_{q,l}\left(t\right)$ to represent the infectious function of position *L* of type *q* at time *t*, which depends on the interaction of all agents (M) at the position at time *t*, and we define it as:(4)$$\lambda_{q,l}\left(t\right)=\frac{1}{n_{l}{}^{a^{q}}}\overset{n_{l}}{\underset{M=1}{\sum }}\left(E_{k}\rho_{k}\beta_{q,k}+I_{k}\psi_{l}C_{k}\rho_{k}\beta_{q,k}\right)$$
where $n_{l}$ represents the number of all agents associated with position type *q*, It represents the weighted ratio between agents in the exposed (E) and infected (I) states and all agents at this position; $a^{q}$ is a scale parameter and $a^{q}\leq 1;E_{k}$ is an indicator function that is equal to 1 if the agent is infectious in the exposed (E) state and 0 otherwise; $\rho_{k}$ represents the infectivity change of our model between agents and $\rho_{k}\geq 0$; $\beta_{q,k}$ represents the transmission rate of COVID−19 and $\beta_{q,k}\geq 0$, The size of $\beta_{q,k}$ depends on the type of Q and the activities of agents under this type, for example, the transmission rate of normal living families is different from that of home isolation; $I_{k}$ indicates whether the agent has symptoms of COVID-19, If there are symptoms, $I_{k}=1$, otherwise, $I_{k}=0$; $\psi_{l}$ is an adjustment parameter for the number of people in workplaces and schools, $\psi_{l}\in \left[0,\;1\right]$ is used to reduce the number of these two kinds of local agents when the epidemic situation appears; $C_{k}$ is a measure of the increased infectivity of agents with symptoms (I) compared to agents exposed (E).

The COVID-19 confirmed intelligence was removed from the model for either recovery (R) or death (D). Our model uses age-based mortality data, the outcome of the treatment of the diseased agent and the current rate of nucleic acid testing and whether the hospital’s medical resources are stretched. The mortality rate of a diseased agent depends on whether the agent is not receiving proper medical care because of medical resource constraints [45,46,47,48,49].We distinguish whether confirmed patients (i.e., agents with symptoms (I)) need ICU treatment or not. Not all agents are requiring ICU treatment will be admitted to ICU, and some agents cannot be admitted to ICU due to the shortage of medical resources, which will have a high mortality rate. In order to more accurately get the COVID-19 disease mortality in the Huangpu District of Shanghai, we use the sick agent can get ICU treatment in ICU treatment need N and cannot get ICU treatment $\bar{N}$ total probability, got sick agent death total probability *P*(D|I) as follows:(5)$$P\left(D|I\right)=P\left(D|N\right)P\left(N\right)+P\left(D|\bar{N}\right)P\left(\bar{N}\right)$$

Through the rearrangement of Equation (5), we can get:(6)$$P\left(D|\bar{N}\right)=\frac{P\left(D|I\right)-P\left(D|N\right)P\left(N\right)}{1-P\left(N\right)}$$

Next, we can deduce the formula of computing needs of P(D|I),P(D|N) and P(N). According to the study of Ferguson et al. [26], the death probability P(D|I) of symptomatic agents can be derived from the available infection fatality rate (IFR). Since IFR is statistically based on serological information, it reflects the mortality rate of infected agents (whether or not the agent has symptoms) [25]. According to the current COVID-19 clinical manifestations, asymptomatic infections will not die, we will IFR redefined as the probability of symptomatic COVID-19 patients is P(I|Cov), so as to calculate P(D|I):(7)$$P\left(D|I\right)=\frac{IFR}{P\left(I|Cov\right)}$$

If diagnosed patients need treatment in the ICU, Equation (6) the death probability P(D|N) is calculated according to whether the current hospital medical resources nervous. Specifically, by the total probability law for the event *T* condition:(8)P(D|N)=1−(1−P(D|N,T))P(T|I)

In Equation (8) P(T|I) is the probability of whether the patient to ICU treatment. Because of IFR used in Equation (6) is the average of the whole COVID-19 popular last time average, so P(T|I) is also used by the average. Probability P(N) of confirming patients receiving ICU treatment and P(H) of all patients requiring ICU treatment is estimated based on the actual situation of COVID-19 in Huangpu District:(9)P(N)=P(N|H)P(H)

Finally, substitute Equations (7)–(9) into Equation (6) to calculate the probability we need. When agents die, they will be removed from the model and no longer contribute to the spread of COVID-19.

### 2.3. Model Calibration

There are many sources for our model parameters: literature data used in other agent models [21,23,24,26], and clinical data of COVID-19 are also from literature [26,50,51,52]. The parameters of length of stay were derived from literature [26], and based on the study of Richardson et al. [50], a linear scaling of coefficients was developed in accordance with Huangpu District of Shanghai. Other parameters are based on literature, official reports and news media reports, and finally adjusted according to model calibration and the actual situation of Huangpu District. Some of the more important parameters (number of pathogens initially infected with COVID-19, rate of nucleic acid testing, agent isolation and reopening of Huangpu District) were set based on official notifications and government measures.

The transmission parameters of COVID-19 in the model are derived from the agent model of Ferguson et al. [26], The transmission parameters used by the research team are those they developed for influenza, which is reasonable because the transmission of COVID-19 is similar to that of respiratory diseases such as influenza. In order to make the transmission rate more realistic, $R_{0}$ can be used to measure, and $R_{0}$ represents the average number of secondary infections directly caused by an infected individual [26]. Based on similar agent models, the $R_{0}$ of COVID-19 is estimated to be 2.4 [26].

## 3. Results

### 3.1. Descriptive Analysis

The agent switched states during the simulation. When the model was initialized, we initialized the whole population in Huangpu District into a susceptible state. A number of agents are then defined at the beginning of the model and assigned states of exposure, and these agents will be tested only after symptoms of COVID-19 appear. When agent after infection, the intelligent experience exposure for a period of time, according to COVID-19 the contagiousness of the virus, we assume that the exposure of the agent at the beginning when the infection is not contagious, but, according to the national health committee of the People’s Republic of China announced that 2019 coronary virus disease of the incubation period is about 10 days, the incubation period is contagious [42], So, we’ll set the agent to be contagious after it’s been exposed for a while. At the same time, the model takes into account asymptomatic situations, where asymptomatic agents are detected in nucleic acid tests of contacts of confirmed patients, and these asymptomatic patients are then quarantined in a centralized manner. To match the specific situation of Shanghai. To simulate the real COVID-19 outbreak in Shanghai’s Huangpu District, we added home quarantine, concentrated quarantine, city-wide lockdown, and three reopening phases I, II, and III.

An agent can be healthy, undergoing nucleic acid testing, or undergoing medical treatment. Similar COVID-19 symptoms of agent (influenza), and has become the exposure of the state of the agent are subject to the nucleic acid detection, we set up two models of nucleic acid testing way, agent can be in the hospital for nucleic acid detection, this way of testing may make the hospital staff and at the same time do other agent infected nucleic acid detection. The other test is to become a close contact (an agent that is already in home quarantine) and wait for a doctor to come in for a test, which we assume is less of a risk of infection. We assume that an agent in a susceptible state will undergo a nucleic acid test some time after the decision to do so (fixation), and the test results will appear some time after the test (fixation). The test results can be positive or negative, with some false positives and false negatives. Once an agent tests positive, the agent is sent to a hospital for treatment, as well as the day’s interchangers (also known as companions). It refers to a period of nucleic acid testing after staying with a confirmed patient for more than 10 min in the same space-time grid (range: 800 m × 800 m). Our model does not apply to any explicit contact tracing, where case testing is achieved in an average sense, where nucleic acid testing on a particular agent depends on evenly distributed random sampling, and where measurement accounting tests performed daily are time-dependent according to reality.

After the test results are available, asymptomatic patients and close contacts of the confirmed patient are quarantined, the confirmed patient is routinely hospitalized, and when the agent becomes sicker, he or she is admitted to ICU. After the onset of COVID-19 symptoms, the treatment status of agents in centralized isolation will change to normal hospitalization or ICU treatment. Agents requiring treatment can change their treatment status based on their treatment status and clinical observations, and the initial treatment of the patient can be obtained from clinical data based on the probability of normal hospitalization and ICU hospitalization, combined with the patient’s age [26,53]. If a patient is admitted to the ICU for treatment, the patient’s recovery status is recalculated based on the ICU mortality rate [26]. After a COVID-19 diagnosis, the agent will switch in the treatment type and finally, all agents infected with COVID-19 will be removed by either a cured COVID-19 patient (R) or a deceased COVID-19 patient (D), and the removed agents will no longer contribute to the transmission of COVID-19. Our model also included agents that have not undergone nucleic acid testing but need ICU treatment, and such agents have a higher probability of death, indicating that the mortality rate is higher without treatment. However, due to China’s policy, this part of the agent will be set to zero. Since the symptoms of COVID-19 disease are similar to those of the common cold or seasonal flu, susceptible agents may experience similar symptoms [54], When they have similar symptoms, they will be isolated until they get a negative nucleic acid test. We assume that the population of agents with influenza is constant, that they will increase the burden of the detection site, and that they will also become infected if they come into contact with an infected person when they are tested. In the model, whether a COVID-19 patient recovers or dies is determined at the time the agent is diagnosed, treated in the ICU and dies in the ICU. After ICU treatment, the intelligence was transferred to normal inpatient treatment after a period of time (fixed) [26]. Any agent confirmed dead is placed in the ICU for a fixed number of days before death [47].

### 3.2. Model Implementation

To demonstrate the usability of the model, we simulated the transmission of COVID-19 in Huangpu District of Shanghai from 8 March to 15 June 2022, from the start of the COVID-19 outbreak to the reopening of Huangpu District. In our calibration of the model, we used the daily number of new COVID-19 cases, the cumulative number of confirmed COVID-19 cases, and the total number of deaths published by the Shanghai Municipal Health Commission. From these data, we extracted the weekly new cases and deaths during the COVID-19 outbreak in Huangpu District. In order to make the model more consistent with the real situation in China, parameters were optimized to change the number of infected people in the initial model, the rate of nucleic acid testing per day, the decrease of the transmission rate in public places during the whole closure of Huangpu District, and the increase of the transmission rate when Huangpu District was reopened in the later stage of the model. As a matter of fact, the number of agents that can be tested daily by nucleic acid varies over time, which can match the number of new confirmed COVID-19 cases per week calculated from the total number of cases during the implementation of the model simulation. We performed 100 simulations (600 steps per simulation (100 days)) of the model, starting with a fixed number of infected agents, and averaging the 100 simulations to match the actual situation of COVID-19 in Huangpu District.

Figure 5 shows the validation results of the model implementation. We compared the output data of the model with the real data of Huangpu District in four aspects (indicators): (a) total number of confirmed COVID-19 cases during the simulation, (b) total number of deaths during the simulation, (c) average number of new cases per week during the simulation, and (d) weekly deaths during the simulation. The total number of confirmed COVID-19 cases is the total number of COVID-19-positive agents detected when the statistical model was run, including false positives and sick agents who died due to medical resource constraints and were not assigned to ICU treatment. The agents of deaths in the models was compared with those reported by the authorities.

As showed in Figure 5, the comparison of the total number of confirmed COVID-19 cases shows that the model running results are consistent with the actual data reported by Shanghai Municipal Health Commission. Model can better predict the number of new cases per day, but, after the comparison, we can observe that the Huangpu District COVID-19 cases of epidemic in the medium term dramatically increases suddenly, compared with the real situation, the model number of cases increase is relatively stable, the patients with us at the start of the epidemic of nucleic acid detection and positive contact tracing, The model provides a simple practical scenario of nucleic acid detection, which may lead to the difference between the operation of the model and real data. It is also related to the centralized and large-scale nucleic acid detection in reality. For the comparison of the average number of deaths per week, the model operational results are consistent with the real data. However, the predicted data is slightly lower than the real data, which may be related to the fact that the recovery time parameter used in the model is less than the real data. As for the number of death cases, we obtained the number of death cases in the whole Shanghai and reduced it to Huangpu District according to the population proportion. Through the comparison of Figure 5, we can find that the number of simulated death cases is consistent with the real data.

Admittedly, mask wearing remains one of the key measures to curb the spread of the pandemic with non-pharmaceutical interventions. To demonstrate the value of our model and the impact of mask wearing on the COVID-19 pandemic, we simulated that all residents in Huangpu District of Shanghai wore masks in public places during the outbreak (i.e., the mask wearing rate was 100%) and maintained 1 m of social distancing. According to the results of the survey on public health behavior conducted by Shanghai Health Promotion Center, 80% of the citizens wear masks when going out in terms of COVID-19 prevention and control. Figure 6 shows a simulation of COVID-19 outbreak in the Huangpu District with a 100% mask wearing rate. In this scenario, we compare the official data with the model output data.

As shown in Figure 6, mask wear rate at 100%, under the situation of the cumulative average model output with the confirmed deaths have declined, but the decline is not obvious, it is associated with the actual situation of China, in the Huangpu District COVID-19 outbreak epidemic period, all the citizens of Shanghai public mask wear rate is 80%, so in this scenario, The output of the model does not decline significantly. However, we still recommend masking recommendations, official guidelines on proper use, and awareness-raising campaigns to shift mask wearing from a purely self-protective mindset to one that responsibly protects the public health of the community.

## 4. Discussion

Common infectious disease models can be divided into SI, SIS, SIR, SIRS and SEIR models according to the characteristics of specific infectious diseases. The SEIR model is suitable for the simulation of COVID-19 virus because it is suitable for the disease with incubation period and certain immunity after cure, which can be applied to four groups of people: susceptible, exposed, sick and recovered. We extended the SEIR model to include death status, nucleic acid testing status, and patient treatment status. Discrete Event simulation (DES) is a method of simulating the behavior and performance of a real life process, facility or system. Our model is based on the SEIR model and combined with discrete event model to form agent model. Therefore, our model has advantages over common models of infectious diseases. First, since the behavior and characteristics of each agent are independent, they can model complex dynamic systems without oversimplifying the rich variation between individuals. Second, because pathogens can be modeled in physical two-dimensional or three-dimensional space, they can better model the geometry of contacts between individuals, which is highly relevant in epidemiology. Third, the randomization of each run makes the statistical variance more pronounced than in the SIR model family, whose smooth curves often misleadingly convey greater certainty than is guaranteed. Fourth, the agent model is well suited for visualization, as shown in Figure 5 and Figure 6. Of course, the shortcomings of the agent model are also prominent. Its computation cost is relatively high, especially when residents maintain a social distance of 1 m in public places under scenario simulation, which results in a long computation time of the model. The computational cost will be reduced, which will be the direction of our next efforts.

During the current COVID-19 pandemic, striking the right balance between safe and normal travel for populations requires the use of effective nucleic acid testing strategies as well as government macro interventions for prevention. During a pandemic, the high volume of nucleic acid testing, unknowns such as the ongoing mutation of the COVID-19 virus, and the complexity and uncertainty of human behaviors all requires the principled use of predictive computing models. So we use the statistical data of Shanghai Huangpu District construction and census data for the seventh time in China, according to the statistical data obtained in the building of the latitude and longitude of the geographical position of Huangpu District in Shanghai modeling, created a series of position of Shanghai Huangpu District (family homes, nursing homes, schools, hospitals, public places, etc.), The one-to-one modeling of residents in Huangpu District based on the census data has provided a more accurate realistic basis for our model, and the prediction results of the model can provide a scientific theoretical basis for effective prevention and control of COVID-19 and public health intervention.

In the model construction, we conducted one-to-one high-precision modeling for residents and geographical buildings in Huangpu District, Shanghai. We implemented technologies related to the COVID-19 agent model: (1) Different nucleic acid testing methods for agents in hospitals and at home; (2) The number of nucleic acid tests performed daily changes with time; (3) Accurate tracking and nucleic acid testing of spatio-temporal intersection population; (4) the burden of nucleic acid testing caused by influenza with the same symptoms as COVID-19 patients; (5) Multiple treatment methods for COVID-19 patients: general hospitalization, intensive care unit treatment, and home isolation after treatment; (6) Separate modeling is conducted for agents working in special places such as schools, hospitals and nursing homes; (7) The blockade and isolation of different places in Huangpu District and the whole Huangpu District and the re-opening of these places; (8) Model building based on literature and various parameters provided by the World Health Organization; (9) Calibrate the model based on COVID-19 data reported by China health Commission and Shanghai Municipal Health Commission. Our model is modelled in Huangpu District. At the district-level scale, our model has been verified and can serve as an information-based decision-making platform for other districts in China.

## 5. Conclusions

Our model has been validated to adjust and reflect on existing COVID-19 intervention strategies and individual health behaviors, and to provide scientific theoretical basis and information-based decision-making tools for effective COVID-19 prevention and control and public health intervention in China. However, there are still some shortcomings in the model. Due to the lack of real crowd movement data in our model, we adopted the Haversine formula in crowd movement. Although we added bus, subway, private car travel and other travel modes, there are still shortcomings in the mobility of agents and random contact of crowds. Model does not include the whole community for nucleic acid detection, while the agent model is going to hospital for nucleic acid detection, doctor visits, and allowed to have similar symptoms of agent for nucleic acid detection, but for the entire community as a whole for nucleic acid detection also shortcomings, these deficiencies can lead to model running late for detection of COVID-19 cases.

Although the model has some shortcomings, the verified model is highly consistent with the real data. We built the COVID-19 epidemic agent risk assessment system that is suitable for the individual epidemiological characteristics of COVID-19 in China, and explored the COVID-19 epidemic intervention and control measures and indicators with universal applicability. The adjustment and reflection of existing COVID-19 intervention strategies and individual health behaviors has provided a scientific theoretical basis and an information-based decision-making tool for effective COVID-19 prevention and control and public health intervention in China.

## Figures and Tables

**Figure 1 ijerph-19-10242-f001:**
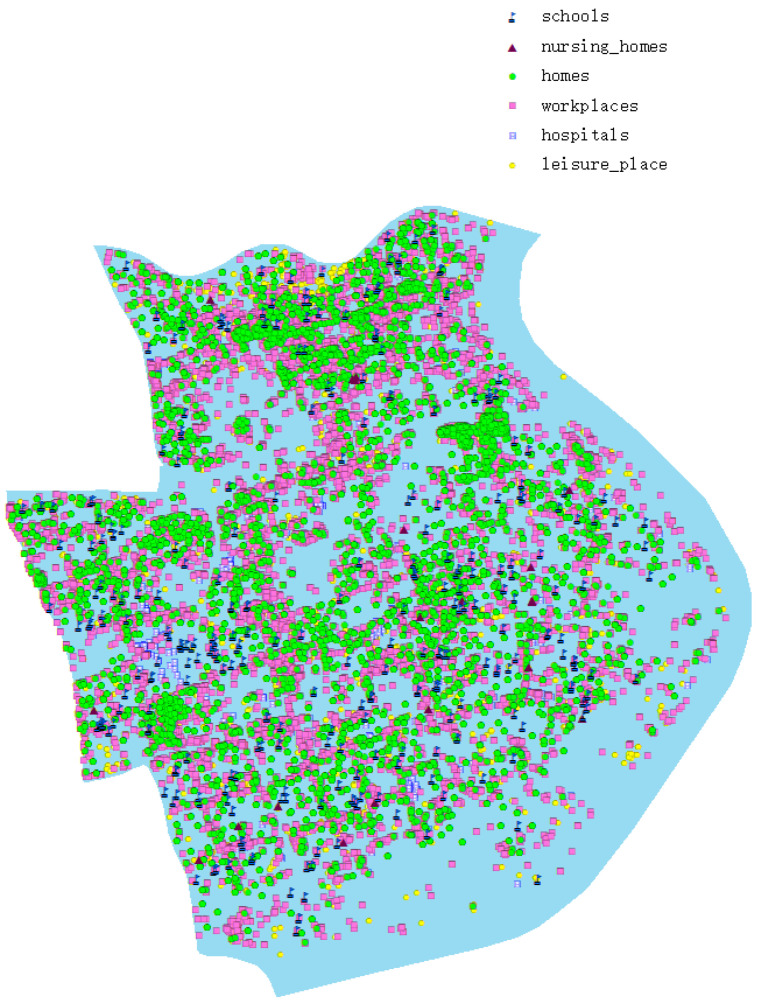
Shanghai Huangpu District map, mainly shows the various geographical buildings in Huangpu District.

**Figure 2 ijerph-19-10242-f002:**
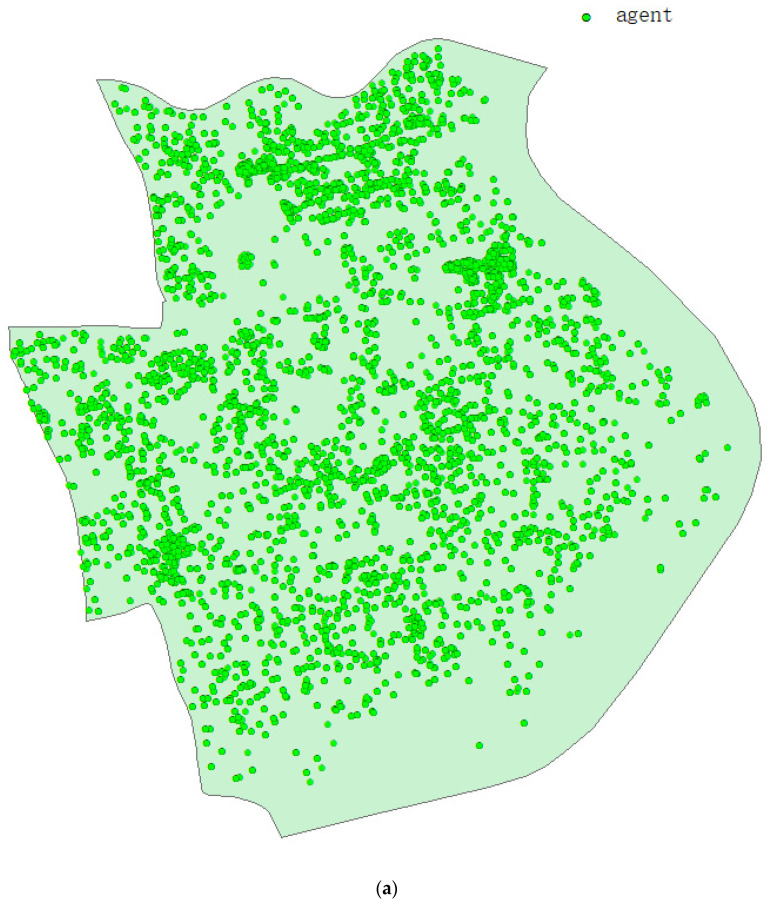
Population characteristics of model creation: (**a**) the initial location of residents in Huangpu District of Shanghai when the model is initialized; (**b**) The population age distribution created for the model.

**Figure 3 ijerph-19-10242-f003:**
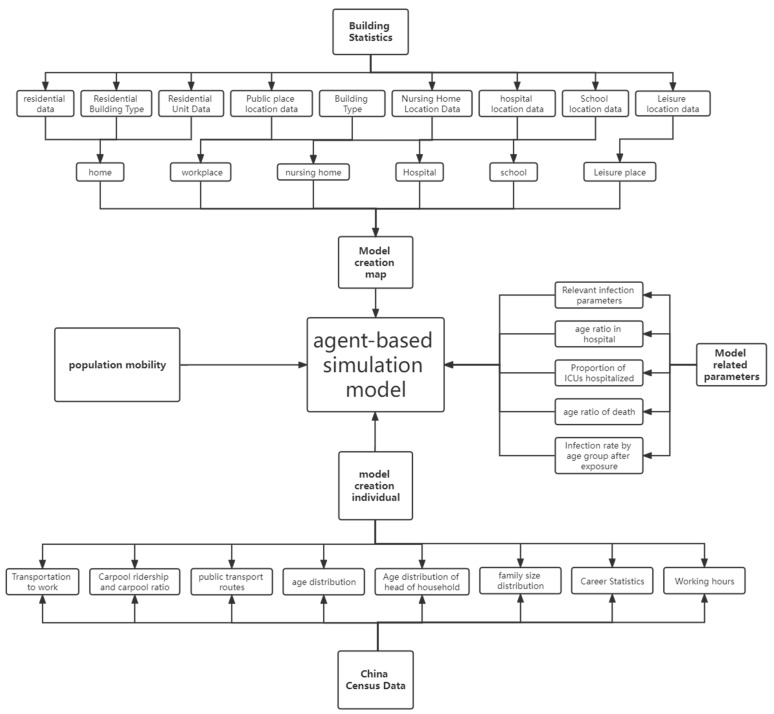
Model architecture diagram.

**Figure 4 ijerph-19-10242-f004:**
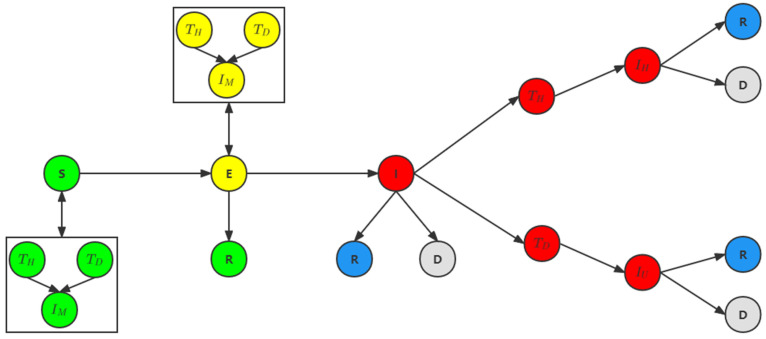
Agent of the state transition: Susceptible (S) for Susceptible state; The Exposed (E) as the exposure condition; Infected (I) is an Infected state; Removed (R) is a state of restoration; Dead (D) refers to the state of death, and the model provides two detection methods, namely, nucleic acid test at the hospital ($T_{H}$) and nucleic acid test at ($T_{D}$) the doctor’s home isolation ($I_{M}$) the model also provides two treatment methods, ordinary hospitalization ($I_{H}$) and ICU treatment ($I_{U}$) required by severe patients.

**Figure 5 ijerph-19-10242-f005:**
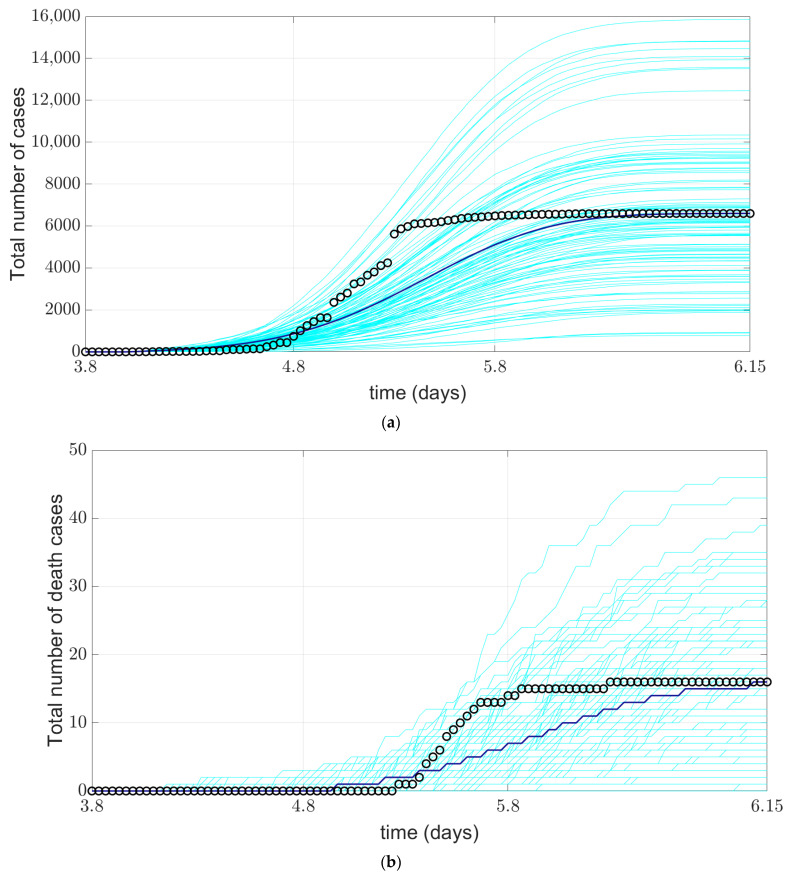
Comparison between the simulated COVID-19 situation in Huangpu District of Shanghai and the official report data, (**a**) comparison of cumulative confirmed cases; (**b**) comparison of cumulative death cases; (**c**) comparison of average weekly new cases; (**d**) comparison of average weekly deaths. The cyan line represents 100 implementations of the model simulation, the blue line represents the average of the model implementation, and o represents the official reported data.

**Figure 6 ijerph-19-10242-f006:**
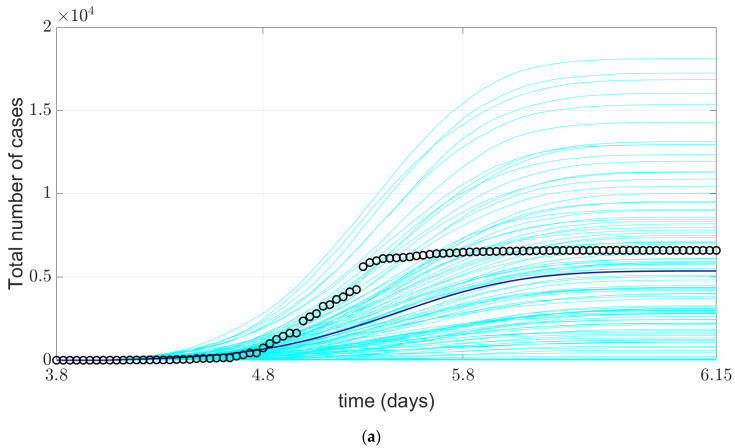
Comparison of simulated COVID-19 situation and official reported data in Huangpu District during scenario simulation (mask wearing and social distancing). (**a**) comparison of cumulative confirmed cases; (**b**) comparison of cumulative death cases. The cyan line in the figure represents the realization of the model simulated 100 times, the blue line represents the average of the model realization, and o represents the official reported data.

## Data Availability

The daily case data of this study were obtained from COVID-19 data provided by the Chinese government, the National Health Commission (http://www.nhc.gov.cn/ (accessed on 15 June 2022)) and the Shanghai Municipal Health Commission (https://wsjkw.sh.gov.cn/ (accessed on 15 June 2022)). The population of Shanghai’s Huangpu District is based on the seventh population census in 2020, China Bureau of Statistics (http://www.stats.gov.cn/ztjc/zdtjgz/zgrkpc/dqcrkpc/index.html (accessed on 10 December 2021)). All kinds of building data from Amap (https://www.amap.com/ (accessed on 10 March 2022)) and Google map (https://www.google.com/maps (accessed on 10 March 2022)). School students and the number of employees and the number of hospital staff and patients in hospital from the Huangpu District official website (https://www.shhuangpu.gov.cn/ (accessed on 10 February 2022)).

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
