# Peer review of "Agent Simulation Model of COVID-19 Epidemic Agent-Based on GIS: A Case Study of Huangpu District, Shanghai"

_ijerph, 2022, doi:10.3390/ijerph191610242_

Round 1
Reviewer 1 Report
The article is very good. But I propose to make two small tweaks:
(1) Graphics are hard to read (especially: font too small and might change it to Arial?).
(2) In the introduction, I suggest adding a few words about the specificity of the Huangpu district (I'm not from China and it's difficult for me to understand all the conditions of this district).
Author Response
Dear Editor and Reviewers
Thank you for your valuable advice. This is very valuable for the improvement of our article. According to your comments, we carefully studied and revised our manuscripts accordingly. We sincerely replied to your comments to ensure that all comments were well handled. It is worth noting that we marked the comments in blue italics and marked the corresponding modifications in the manuscript in red font. The following is our response to each comment:
Comment 1:
Graphics are hard to read (especially: font too small and might change it to Arial?).
Response: Thank you for your comment. We have made adjustments to the graphics in the manuscript. You can check it in the revised manuscript.
Comment 2:
In the introduction, I suggest adding a few words about the specificity of the Huangpu district (I'm not from China and it's difficult for me to understand all the conditions of this district).
Response: Thank you for your comment. We have carried on the supplementary explanation to the particularity of Huangpu District. The above description can be found in Lines 26-27.
Please see the attachment

Reviewer 2 Report
The manuscript is well structured and written, however there are several concerns regarding the relevance and novelty of the content.
There are several studies on space-time simulation methods based on the available big data and different models proposed for different diseases, including dengue and COVID-19. There is a study of COVID-19 that includes Huangpu district (the study region proposed by the authors), among others.
Zhou, S., Zhou, S., Zheng, Z., Lu, J., & Song, T. (2022). Risk assessment for precise intervention of COVID-19 epidemic based on available big data and spatio-temporal simulation method: Empirical evidence from different public places in Guangzhou, China. Applied Geography, 143, 102702.
Sánchez-Hernández, D., Aguirre-Salado, C. A., Sánchez-Díaz, G., Aguirre-Salado, A. I., Soubervielle-Montalvo, C., Reyes-Cárdenas, O., ... & Santana-Juárez, M. V. (2021). Modeling spatial pattern of dengue in North Central Mexico using survey data and logistic regression. International Journal of Environmental Health Research, 31(7), 872-888.
On line 65 such references can be included, as well as a brief paragraph related to these works.
Authors must compare their model with others in the state of the art, or claim the new in contrast to different works in the state of the art.
I attach a PDF document with some comments and doubts about the content, and suggestions related to the use of English throughout the manuscript.

Author Response
Dear Editor and Reviewers
Thank you for your valuable advice. This is very valuable for the improvement of our article. According to your comments, we carefully studied and revised our manuscripts accordingly. We sincerely replied to your comments to ensure that all comments were well handled. It is worth noting that we marked the comments in blue italics and marked the corresponding modifications in the manuscript in red font. The following is our response to each comment:
Comment 1:
On line 65 such references can be included, as well as a brief paragraph related to these works.
Response: Thank you for your comment. We have made changes to the manuscript. You can check it in the revised draft. We will mark the changes in red. The above description can be found in Lines 127-133. Risk assessment for precise intervention of COVID-19 epidemic based on available big data and spatio-temporal simulation method: Empirical evidence from different public places in Guangzhou, China. The Huangpu District mentioned in this paper is not consistent with the one we studied. Our research area is Huangpu District of Shanghai, while the one mentioned in this paper is Huangpu District of Guangzhou.
Comment 2:
Authors must compare their model with others in the state of the art, or claim the new in contrast to different works in the state of the art.
Response: Thank you for your comment. We have made changes to the manuscript. Our model is compared with other epidemic models. You can check it in the revised draft. We will mark the changes in red. The above description can be found in Lines 538-559.
Comment 3:
I attach a PDF document with some comments and doubts about the content, and suggestions related to the use of English throughout the manuscript.
Response: Thank you for your comment. We have made changes to the manuscript. We have improved on some of your comments and doubts about the content, as well as suggestions related to the use of English throughout the manuscript. You can check it in the revised draft. We will mark the changes in red.
Please see the attachment

Reviewer 3 Report
The manuscript explores the ABM approach to simulate the spread of SARS-COV2 in specific cities in China. The model is very detailed and demonstrates the usefulness of ABM in predicting the dynamics of infectious disease spread.
A few major comments are:
1) There are different prediction models including SIR and discrete event simulation models. The authors should discuss advantages and disadvantages of ABM in comparison to other well established model in detail. In particular, ABM is known to be computationally more expensive and sensitive to original assumptions related to behavior when compared to other models. Authors should at least provide some insights into the computational time and sensitivities to the assumptions made.
2) The authors claims the scenario analysis as one of the merits for performing ABM. However, the result section does not provide any results of specific scenario analyses. The study should perform at least some key scenario analyses to add some contexts to the ABM model developed. If this is not possible or irrelevant for some reasons, please address this in the discussion/limitation section.
Other minor comments include:
1) The legends of the figures are too small.
2) In the paragraph under Figure 2, there is a typo. Chin[41]a, should be China [41].
3) Please spell out SEIR when used for the first time.
4) Can you please define delta t further? Is this defined with some specific time interval, e.g., 1 day? Or is it defined based on an event like in a discrete event simulation model?
After these comments are addressed, I think that this manuscript could be a good publication for the journal to demonstrate the usefulness of ABM.
Author Response
Dear Editor and Reviewers
Thank you for your valuable advice. This is very valuable for the improvement of our article. According to your comments, we carefully studied and revised our manuscripts accordingly. We sincerely replied to your comments to ensure that all comments were well handled. It is worth noting that we marked the comments in blue italics and marked the corresponding modifications in the manuscript in red font. The following is our response to each comment:
Comment 1:
There are different prediction models including SIR and discrete event simulation models. The authors should discuss advantages and disadvantages of ABM in comparison to other well established model in detail. In particular, ABM is known to be computationally more expensive and sensitive to original assumptions related to behavior when compared to other models. Authors should at least provide some insights into the computational time and sensitivities to the assumptions made.
Response: Thank you for your comment. We have made changes to the manuscript. Our model is compared with the latest models. You can check it in the revised draft. We will mark the changes in red. The above description can be found in Lines 538-559.
Comment 2:
The authors claims the scenario analysis as one of the merits for performing ABM. However, the result section does not provide any results of specific scenario analyses. The study should perform at least some key scenario analyses to add some contexts to the ABM model developed. If this is not possible or irrelevant for some reasons, please address this in the
Response: Thank you for your comment. We have made changes to the manuscript. We have added a scenario analysis of mask wearing rate to the results section and marked it in red, which you can view in the revised version. The above description can be found in Lines 510-536.
Comment 3:
Other minor comments
Response: Thank you for your comment. As for the other minor comments, we have also made changes in the manuscript and marked them in red. You can check them in the revised manuscript.
Please see the attachment

Round 2
Reviewer 2 Report
The manuscript has been improved by the authors and now their contribution is clearer and the quality of the presentation is better.
Author Response
Dear Editor and Reviewers
Thank you very much for your valuable comments, which have greatly helped us to revise the paper. Thank you again and wish you and your family good health during this pandemic.

Reviewer 3 Report
The authors have done a nice work to incorporate all comments that I provided. Thanks so much for the diligent work and it is now reads very well.
Author Response

(The authors gave the same response as above.)
